# High Radial Artery Puncture Reduces CRPS Risk for Women: A Retrospective Case Series

**DOI:** 10.3390/jcm14175937

**Published:** 2025-08-22

**Authors:** Takehiro Hashikata, Masahiko Shibuya, Yoshiaki Shintani, Koichi Miyazaki, Yuji Okuno

**Affiliations:** 1Department of Radiology, Musculoskeletal Intervention Center, Okuno Clinic, Tokyo 106-0032, Japan; shibuya@okuno-y-clinic.com (M.S.); 2010kasayoki@gmail.com (Y.S.); miyazaki@okuno-y-clinic.com (K.M.); okuno@okuno-y-clinic.com (Y.O.); 2Department of Cardiovascular Medicine, Kitasato University School of Medicine, Sagamihara 252-0324, Japan; 3Department of Cardiology, Ageo Central General Hospital, Ageo 362-8588, Japan

**Keywords:** radial artery access, high radial artery puncture, complex regional pain syndrome, intravascular intervention, women

## Abstract

**Background/Objectives:** Radial artery access (RAA) is widely used for catheter-based procedures due to its safety and convenience, but complex regional pain syndrome (CRPS) remains a rare, underrecognized complication—particularly in women. CRPS manifests as prolonged, severe pain and autonomic symptoms, often associated with nerve irritation near the carpal tunnel. This study aimed to evaluate whether modifying the puncture site to a high radial artery puncture (HRAP) reduces the risk of CRPS in patients undergoing transarterial micro-embolization (TAME) for frozen shoulder. **Methods:** We retrospectively reviewed 97 patients (47 women and 50 men) who underwent transarterial micro-embolization (TAME) via conventional RAA for frozen shoulder between February and June 2019. The occurrence of CRPS and vascular complications was recorded. All punctures were ultrasound-guided. **Results:** Among women treated via conventional RAA, five developed CRPS and one had radial artery occlusion. CRPS symptoms included intense puncture site pain (mean duration was 47 days), which severely impaired daily function. No complications occurred in men. Following the adoption of HRAP, no cases of CRPS, prolonged pain, or vascular complications were observed in the consecutive 101 women treated. **Conclusions:** Our findings suggest HRAP reduces CRPS risk by avoiding superficial nerve branches and targeting deeper arterial segments with fewer sensory structures. This ultrasound-guided modification is simple, does not require additional training, and may be widely applicable in both musculoskeletal and cardiovascular interventions. HRAP may help minimize neuropathic complications in broader patient populations.

## 1. Introduction

Radial artery access (RAA) is widely used for catheter-based interventions due to its superior safety profile and patient comfort compared to femoral access. In the field of coronary intervention, RAA has been associated with lower rates of access-site bleeding complications, faster patient mobilization, and improved overall clinical outcomes. These advantages have made RAA the default access site in many cardiovascular centers worldwide.

Despite its safety and convenience, RAA is not without complications. RAA remains a favored choice for catheter-based procedures due to its safety and convenience, but it is not free of complications. Radial artery occlusion, which is the most frequently reported complication, occurs in approximately 4–6% of cases based on large meta-analyses [1,2]. Clinically significant hematoma occurs in 1.2–2.6% of procedures [3]. Pseudoaneurysm is rare, with an incidence of 0.08% [4]. Compartment syndrome is exceedingly uncommon, occurring in 0.004–0.05% of cases [3]. While nerve injury and complex regional pain syndrome (CRPS) following RAA are rare and mostly described in case reports, they remain underrecognized and may lead to prolonged morbidity. CRPS manifests with a wide spectrum of symptoms, including persistent burning or throbbing pain, allodynia (pain due to normally non-painful stimuli), temperature hypersensitivity, skin color or texture changes, and signs of sympathetic overactivity such as swelling and abnormal sweating in the affected limb [5,6]. The severity and duration of symptoms can vary widely among individuals, often leading to significant functional impairment and reduced quality of life.

CRPS is typically classified into two types: Type I occurs without identifiable nerve damage, often after soft tissue injury or immobilization; Type II is associated with direct peripheral nerve injury. In the context of RAA, anatomical proximity to the carpal tunnel may predispose patients to carpal tunnel syndrome, further complicating the clinical picture and potentially prolonging recovery [7].

Transarterial micro-embolization (TAME) of abnormal neovessels is an emerging, minimally invasive interventional technique that has shown promising results in reducing chronic joint pain, particularly in patients with conditions such as knee osteoarthritis and frozen shoulder [8]. TAME offers a targeted, low-invasive approach that can shorten the duration of pain symptoms. In upper limb interventions, including shoulder and elbow treatments, RAA is frequently used due to its accessibility.

However, in our clinical experience, a small subset of female patients undergoing TAME via conventional radial artery access developed CRPS, which prompted us to reconsider our access strategy. By modifying the puncture site to a higher location on the radial artery, we successfully eliminated further occurrences of CRPS. In this report, we present a case series demonstrating that high radial artery puncture may reduce the risk of CRPS in female patients. Furthermore, given that RAA is the predominant approach in cardiovascular interventions worldwide, we also discuss the potential applicability of this technique in the field of cardiac catheterization, where it may offer similar benefits in reducing access-site related complications.

## 2. Case Series

All participants agreed to take part in this research and provided written consent in accordance with the ethical guidelines approved by the Ethics Committee of Yuyu Medical Corporation Ethics Review Committee (Approval Code: OC 2025-001).

From February to June 2019, our institution performed ultrasound-guided conventional radial artery punctures (within 2 cm proximal to the radial styloid process, where the radial artery is most palpable) for TAME in frozen shoulder patients using a 3Fr sheath. Inclusion criteria were patients aged > 20 and <80 years, with a clinical diagnosis of adhesive capsulitis or symptomatic rotator cuff tears, who had undergone at least 3 months of conservative therapy under orthopedic supervision without improvement, and with stable major organ function. Exclusion criteria included rheumatoid arthritis, neoplasm, severe arteriosclerosis, pregnancy, dementia, joint instability, iodine allergy, severe cardiac or cerebrovascular disease, neurological disease, uncontrolled diabetes mellitus, severe infection, bleeding tendency, and pain due to cervical or lumbar spine pathology. Hemostasis was achieved with a 9 mm-thick compression pad and elastic adhesive tape, followed by a hemostatic band for 20 min, then maintained with the adhesive pad alone for another 20 min.

During this period, 97 patients (47 women and 50 men) underwent the procedure. Among women, 6 (13%) developed radial artery-related complications, including 5 with CRPS and 1 with radial artery occlusion, while no such injuries occurred in men. CRPS patients experienced severe pain at the puncture site immediately post-procedure, persisting for over a month in all cases (mean: 47 days, range: 30–90 days). Pain severity (Numerical Rating Scale ≥ 8) impaired daily activities, including work and self-care.

Among the six women who developed CRPS following conventional radial artery access for TAME, all presented with idiopathic, refractory frozen shoulder and shared clinical features of heightened pain sensitivity and poor response to conservative orthopedic management. Most patients reported immediate and intense pain at the puncture site following sheath removal, with symptoms persisting for 1–3 months. Two patients developed notable vascular complications: one experienced puncture site discoloration and transient radial artery occlusion confirmed by ultrasound (Figure 1), while another developed Raynaud-like symptoms, including digital pallor and coldness, which resolved within two months. In three cases, neuropathic symptoms extended beyond localized pain, manifesting as paresthesia in the hand or digits; one patient described numbness of the thumb lasting over six months. Another patient, with a prior history of carpal tunnel syndrome, reported symptom recurrence post-procedure, suggesting possible anatomical vulnerability. Procedural pain hypersensitivity was observed in all six patients, with one case requiring intra-procedural analgesia with pentazocine. All patients exhibited disproportionate pain responses during and after the procedure, suggesting a shared underlying central sensitization or heightened pain perception. Moreover, their clinical course indicated a common pattern of psychological hypervigilance and treatment resistance, likely rooted in mental stress or anxiety, which may have contributed to both the onset and persistence of CRPS symptoms. Shoulder pain improvement following TAME varied; while one patient experienced a marked reduction in numerical rating scale score from 10 to 5, others reported limited or no functional recovery, indicating that CRPS may have masked or interfered with expected therapeutic outcomes.

Based on the hypothesis that puncture sites close to the carpal tunnel increase CRPS risk, we modified the site, shifting it at least 5 cm proximally to the radial styloid process (high radial artery puncture: HRAP) (Figure 2). Between June and December 2020, TAME was performed via HRAP on 101 female patients with frozen shoulder, and no cases of CRPS, prolonged puncture site pain or arterial occlusion were observed.

## 3. Discussion

Our results suggest that HRAP notably reduces the risk of CRPS compared to the conventional radial approach in patients undergoing transarterial micro-embolization (TAME) for frozen shoulder. Although CRPS is generally regarded as a very rare complication of RAA, we suspect that its true frequency is underestimated due to under-recognition and misdiagnosis. In daily interventional cardiology practice, we occasionally encounter CRPS-like cases at a frequency not negligible compared with that of radial artery occlusion, albeit with a wide spectrum of severity. In the present cohort, CRPS occurred in 6 of 97 patients overall (6 of 47 female patients), representing a much higher incidence than that reported in the cardiovascular literature. We believe this may be attributable to the abnormal pain perception inherent to the affected side in frozen shoulder, combined with the prolonged course of pain symptoms, which likely promotes central sensitization [9]. These patients often presented significant challenges in subsequent clinical management. Importantly, after the adoption of HRAP, no CRPS cases were observed among over 100 female cases, suggesting that this simple modification may offer a practical means of mitigating this complication in high-risk populations.

The RAA has gained wide acceptance in cardiovascular medicine, largely due to its favorable safety profile, lower incidence of access-site bleeding, early ambulation, and overall improved outcomes compared with the femoral approach. Multicenter randomized trials have demonstrated that RA significantly reduces major bleeding and mortality in patients with coronary artery disease [10,11]. These benefits have led to guideline-endorsed recommendations promoting the radial approach as the default access for coronary procedures.

In recent years, the distal radial approach (DRA)—accessing the artery at the anatomical snuffbox—has been proposed as an alternative to further reduce complications such as radial artery occlusion. Studies have shown that DRA may preserve the proximal radial artery for future use and offers cosmetic advantages [12]. However, DRA has certain limitations: it requires a steep learning curve, smaller-caliber vessels may preclude the use of larger sheaths, and puncture success rates are generally lower, particularly in female or elderly patients with weak pulses [13]. By contrast, HRAP provides a straightforward modification of the conventional approach. It does not require additional equipment or training, and it can be readily adopted by operators familiar with standard RAA. The puncture is simply shifted proximally by several centimeters, which offers multiple advantages: the radial artery at this level lies deeper and away from superficial nerve branches, notably the superficial branch of the radial nerve, thus reducing the risk of neurogenic pain syndromes such as CRPS [14]. Importantly, HRAP is applicable in nearly all cases where conventional RAA is possible, making it highly versatile.

Although our report focuses on patients with frozen shoulder, this anatomical modification may be of broader significance. In real-world clinical practice, radial artery access is far more commonly used in cardiovascular interventions than in musculoskeletal procedures. Among these, a growing population of patients with ischemia with non-obstructive coronary arteries (INOCA) and myocardial infarction with non-obstructive coronary arteries (MINOCA) has been recognized [15]. These conditions are more prevalent in women and often involve abnormal pain perception, central sensitization, and autonomic dysregulation—features strikingly similar to those seen in frozen shoulder patients, particularly middle-aged women with heightened pain sensitivity and psychological stressors. For example, nearly two-thirds of women with INOCA exhibit central sensitization, which correlates with depression and elevated pain responses [16]. Mental stress, depression, and anxiety are more prevalent among INOCA/MINOCA patients and are closely linked to microvascular dysfunction and pain amplification [17]. Thus, the demographic and pathophysiological resemblance between our frozen shoulder cohort and INOCA/MINOCA patients suggests HRAP could effectively reduce access-related neuropathic pain in both groups.

According to the updated European Society of Cardiology guidelines, invasive testing—such as functional hemodynamic assessment and acetylcholine provocation—is strongly recommended for diagnosing INOCA/MINOCA [18]. As these procedures increase, it is essential to optimize vascular access to minimize complications, especially in women predisposed to pain syndromes. Given the overlap in pain sensitization mechanisms between patients with frozen shoulder and INOCA/MINOCA populations, HRAP may represent a safer approach that reduces nerve-related sequelae following radial access.

Taken together, HRAP represents a practical and effective refinement of radial access that could improve the safety of both musculoskeletal and cardiovascular interventions, particularly for patients predisposed to neuropathic pain syndromes. Further prospective studies in broader populations, especially in coronary catheterization settings, are warranted to validate its generalizability and long-term benefits.

One limitation of our study is its retrospective observational design, which introduces potential selection bias and limits causal inferences. Additionally, our cohort consisted of adhesive capsulitis patients, a population with heightened pain signaling and possibly increased vascular reactivity, which may overestimate HRAP benefits. Furthermore, all procedures were performed using a 3Fr sheath, differing from the larger sizes (4–6Fr) typically used in trans-radial interventions. However, HRAP remains viable even with larger catheters and may provide superior complication prevention.

## 4. Conclusions

HRAP notably reduced the incidence of CRPS in our female patients undergoing TAME for frozen shoulder. This simple modification of the puncture site—shifting proximally by approximately 3–4 cm—may avoid superficial nerve branches and minimize nerve trauma. HRAP was performed safely using standard ultrasound guidance without requiring additional equipment or training. No vascular complications or prolonged post-procedural pain were observed in the HRAP group. Given the far greater volume of RAA procedures in the cardiovascular field, HRAP may have broader relevance in reducing access-site neuropathic complications in that domain as well.

## Figures and Tables

**Figure 1 jcm-14-05937-f001:**
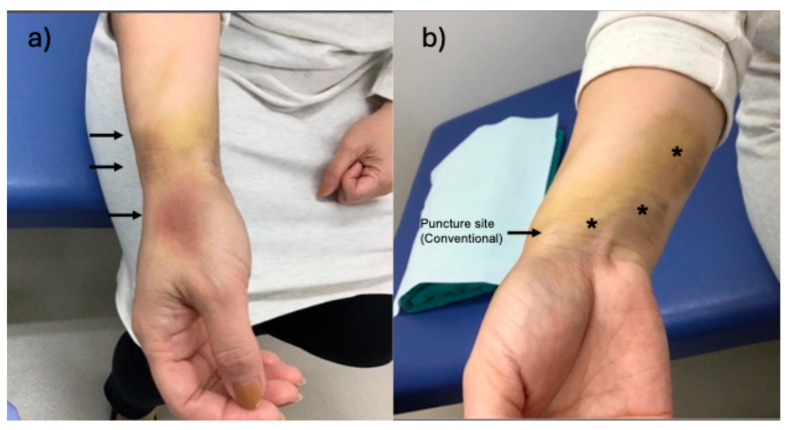
A representative case: A 47-year-old woman with idiopathic frozen shoulder for 2 years received the TAME. TAME was conducted via conventional radial artery access. Immediately after the procedure, she revealed severe pain at the puncture site, which appeared discolored the following day. At the two-week follow-up after the procedure, the radial artery was no longer palpable and was confirmed as occluded by ultrasound. (**a**,**b**) show photographs of the wrist from different angles taken during this visit. The dorsum of the thumb showed redness, numbness, and pain (arrows in (**a**)), while a subcutaneous hematoma was visible on the dorsal wrist (asterisks in (**b**)). One month later, radial pulse returned, and local wrist symptoms gradually resolved. However, she became increasingly fixated on persistent shoulder pain, which did not improve following TAME, and was eventually lost to follow-up one year later.

**Figure 2 jcm-14-05937-f002:**
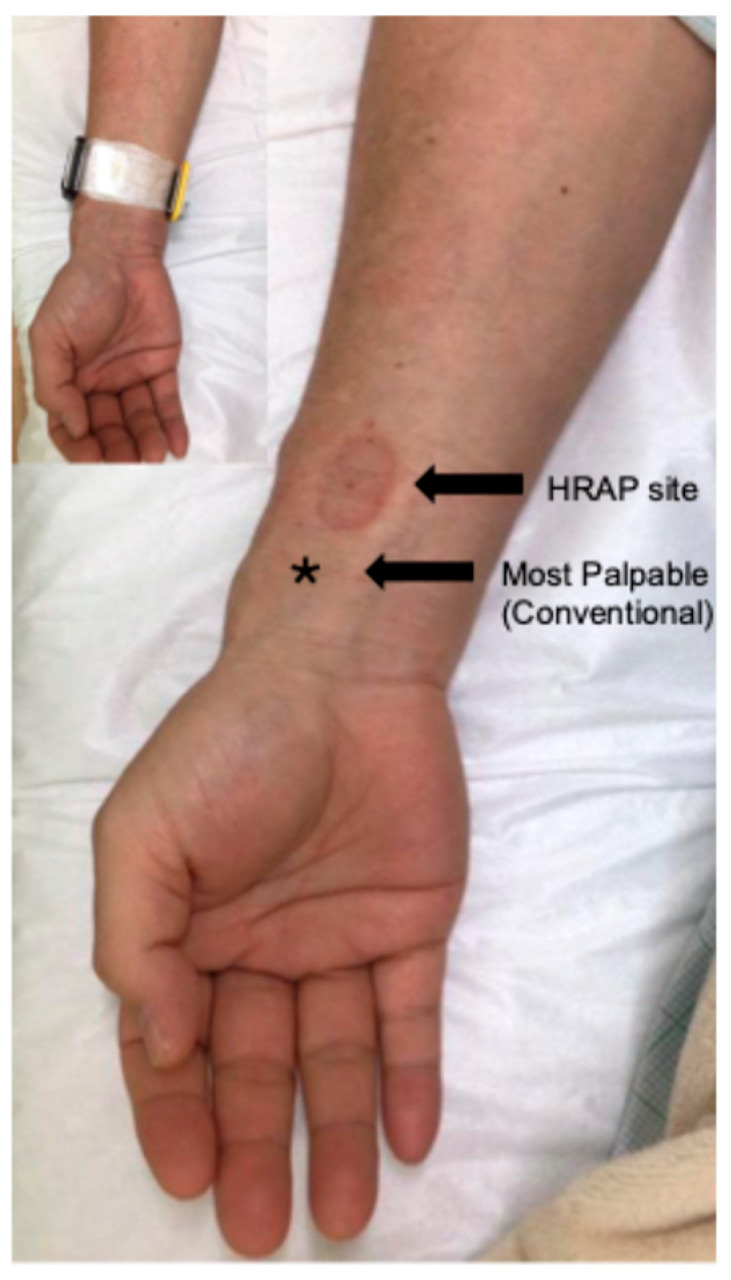
The location of high radial artery puncture. The conventional radial artery puncture site is the most palpable location (star), typically found 2 cm proximal to the wrist. The HRAP is performed by intentionally puncturing a further 3–4 cm proximal to the most palpable point.

## Data Availability

The data supporting the findings of this study are not publicly available due to ethical restrictions. Access to the data can be obtained upon reasonable request to the corresponding author.

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
