# Peer review of "High Radial Artery Puncture Reduces CRPS Risk for Women: A Retrospective Case Series"

_jcm, 2025, doi:10.3390/jcm14175937_

Round 1

Reviewer 1 Report

Comments and Suggestions for Authors

Dear authors of the paper:

I have some constructive comments on your work.

Introduction

Generally speaking, bibliographic citations are missing from the statements you make.

Case Studies.

Did you perform statistical testing?

Discussion

At the beginning of the discussion, you mention that your data show a "significant" risk reduction...

However, you did not use statistical testing. Therefore, I suggest omitting the word "significant."

Conclusion

Please conclude your work conclusively. It seems to me that you are continuing to discuss the issue.

Author Response

Comment 1:  Introduction Generally speaking, bibliographic citations are missing from the statements you make.

Response 1: Thank you for your comment. We have re-checked the introduction and ensured that key statements are supported by appropriate references.

Comment 2: Case Studies.  Did you perform statistical testing?

Response 2: As this study is a descriptive retrospective case series, no statistical testing was performed. 

Comment 3: Discussion  At the beginning of the discussion, you mention that your data show a "significant" risk reduction... However, you did not use statistical testing. Therefore, I suggest omitting the word "significant."

Response 3: We agree with your suggestion and have replaced the term “significantly” with “notably” to avoid implying statistical testing.

Comment 4: Conclusion  Please conclude your work conclusively. It seems to me that you are continuing to discuss the issue.

Response 4: Thank you for pointing that out. We have deleted the sentence "Although this report focuses on patients with frozen shoulder, similar clinical backgrounds—including age, sex, and heightened pain sensitivity—are shared by patients with INOCA/MINOCA, a common condition in cardiovascular medicine." in the conclusion section to make it more concise.

Reviewer 2 Report

Comments and Suggestions for Authors

Dear Editor, dear Authors,

Thank you for the opportunity to review this manuscript.

I appreciate the authors’ efforts in investigating the occurrence of complex regional pain syndrome (CRPS) following conventional radial artery access (RAA) in women undergoing transarterial micro-embolization and in assessing whether shifting the puncture site proximally (high radial artery puncture, HRAP) could reduce this complication. Their findings suggest that HRAP, performed ≥5 cm proximal to the radial styloid, eliminates CRPS cases by avoiding superficial nerve branches, offering a simple, ultrasound-guided modification with potential applicability in other catheter-based interventions.

To develop this study, the authors retrospectively reviewed 97 patients (47 women and 50 men) who underwent the procedure between February and June 2019. Among the female patients, 6 (13%) developed radial artery–related complications, including 5 cases of complex regional pain syndrome (CRPS) and 1 case of radial artery occlusion, whereas no such complications were observed in male patients.

I would like to commend the authors for their efforts and the research conducted in preparing this manuscript. The topic is relevant and of potential interest to the readership. Nevertheless, the review of the existing literature in relation to the present study appears somewhat limited. A more comprehensive literature background would strengthen the manuscript and help prevent the subject from being perceived as insufficiently supported. I would therefore encourage the authors to enrich the Introduction and Discussion sections by including additional, up-to-date references.

In the second paragraph of the Introduction, complex regional pain syndrome (CRPS) is presented as a “rare but unrecognized” complication of radial artery access (RAA). To enhance clarity and contextual understanding, it would be useful for the authors to briefly summarise the main complications of RAA and indicate where CRPS stands in terms of frequency relative to these other complications.

In Section 2, “Case Series,” I recommend centering Figures 1 and 2 for improved visual presentation. Additionally, for Figure 1, please ensure that panels (a) and (b) are clearly labeled and that their specific features or differences are explained in the figure legend. It would also be helpful to indicate, within the legend, what the arrows in Figure 1a are pointing to.

Given the retrospective nature and case series design of the study, I would suggest that the authors adopt a more cautious approach in using the term “demonstrates.” Considering the position of this type of study in the hierarchy of evidence, and since this type of research cannot support strong causal inferences, it would be more appropriate to use phrasing such as “the study suggests.”

Finally, I also recommend that the authors ensure that references are placed within the sentence rather than outside of it. For example, instead of writing “. [1,2]”, it would be preferable to write “[1,2].”

I hope this review proves constructive and contributes to the improvement of the manuscript. The suggestions provided are not mandatory; the authors are encouraged to implement those they find reasonable and valuable.

I appreciate the effort the authors have invested in the research and preparation of this work.

With respect and consideration,

Reviewer.

Author Response

Comment 1: I would like to commend the authors for their efforts and the research conducted in preparing this manuscript. The topic is relevant and of potential interest to the readership. Nevertheless, the review of the existing literature in relation to the present study appears somewhat limited. A more comprehensive literature background would strengthen the manuscript and help prevent the subject from being perceived as insufficiently supported. I would therefore encourage the authors to enrich the Introduction and Discussion sections by including additional, up-to-date references.
In the second paragraph of the Introduction, complex regional pain syndrome (CRPS) is presented as a “rare but unrecognized” complication of radial artery access (RAA). To enhance clarity and contextual understanding, it would be useful for the authors to briefly summarise the main complications of RAA and indicate where CRPS stands in terms of frequency relative to these other complications.

Response 1: Thank you very much for your constructive ideas. We totally agree with you, and have added information on the complications of RAA reported to date and how this case series relates to them in the introduction and discussion sections to help readers better understand the article.

Added sentences in Introduction: "RAA remains a favored choice for catheter-based procedures due to its safety and convenience, but it is not free of complications. Radial artery occlusion, which is the most frequently reported complication, occurs in approximately 4–6 % of cases based on large meta-analyses [1,2]. Clinically significant hematoma occurs in 1.2–2.6 % of procedures [3]. Pseudoaneurysm is rare, with an incidence of 0.08 % [4]. Compartment syndrome is exceedingly uncommon, occurring in 0.004–0.05 % of cases [3]. While nerve injury and complex regional pain syndrome (CRPS) following RAA are rare and mostly described in case reports, they remain underrecognized and may lead to prolonged morbidity. "

Added sentences in Discussion: "Although CRPS is generally regarded as a very rare complication of RAA, we suspect that its true frequency is underestimated due to underrecognition and misdiagnosis. In daily interventional cardiology practice, we occasionally encounter CRPS like cases at a frequency not negligible compared with that of radial artery occlusion, albeit with a wide spectrum of severity. In the present cohort, CRPS occurred in 6 of 97 patients overall (6 of 47 female patients), representing a much higher incidence than that reported in the cardiovascular literature. We believe this may be attributable to the abnormal pain perception inherent to the affected side in frozen shoulder, combined with the prolonged course of pain symptoms, which likely promotes central sensitization [9]. These patients often presented significant challenges in subsequent clinical management. Importantly, after the adoption of HRAP, no CRPS cases among over 100 female cases were observed, suggesting that this simple modification may offer a practical means of mitigating this complication in high-risk populations."

Comment 2: In Section 2, “Case Series,” I recommend centering Figures 1 and 2 for improved visual presentation. Additionally, for Figure 1, please ensure that panels (a) and (b) are clearly labeled and that their specific features or differences are explained in the figure legend. It would also be helpful to indicate, within the legend, what the arrows in Figure 1a are pointing to.

Response 2: Thank you for your suggestion. Based on your valuable comments, we have moved Figures 1 and 2 to the center. In addition, we have labeled each photo in Figure 1 and added explanations to make it easier for readers to understand.

Comment 3: Given the retrospective nature and case series design of the study, I would suggest that the authors adopt a more cautious approach in using the term “demonstrates.” Considering the position of this type of study in the hierarchy of evidence, and since this type of research cannot support strong causal inferences, it would be more appropriate to use phrasing such as “the study suggests.”

Response 3: Thank you for pointing that out. We agree with you, but unfortunately, we were unable to find any instances of the word “demonstrate” in the text that were similar to those pointed out by the reviewer, therefore we were unable to make any corrections. There was one instance of “mulcenter randomized trails have demonstrated...,” but we do not believe that this needs to be changed, so we have left it as is.

Comment 4: Finally, I also recommend that the authors ensure that references are placed within the sentence rather than outside of it. For example, instead of writing “. [1,2]”, it would be preferable to write “[1,2].”

Response 4: Thank you very much for pointing that out. We have made the necessary corrections as you suggested.

Reviewer 3 Report

Comments and Suggestions for Authors

The presented work is devoted to an important clinical problem — complications during radial artery access (RAA). Complex regional pain syndrome (CRPS) is a serious complication that requires close attention, especially in the female population. The scientific novelty of the work lies in the proposal of a modified radial artery puncture (HRAP) technique that reduces the risk of CRPS. The relevance of the study is due to the growing frequency of radial access in modern interventional medicine.
The authors conducted a retrospective study followed by a prospective analysis of the modified technique. The study design includes: analysis of 97 patients (47 women, 50 men), description of complications, implementation of the modified HRAP technique. The study methodology is justified, the sample is representative for assessing the phenomena under study.

Remarks.
1. The section on patients does not contain information on the approval of the study by the ethics committee, as well as on the signing of informed consent by patients to participate in the study.
2. The criteria for inclusion and exclusion of patients are not specified.
3. The text contains information from the article formatting template that needs to be removed (page 5, end of the discussion section).

Author Response

Comment 1. The section on patients does not contain information on the approval of the study by the ethics committee, as well as on the signing of informed consent by patients to participate in the study.

Response 1: Thank you for your indication. We have included the information regarding our ethics committee approval and patient informed consent in the beginning of Case Series section of the revised manuscript.

Comment 2. The criteria for inclusion and exclusion of patients are not specified.

Response 2: Thank you for your adequate suggestion. We have added a detailed description of the inclusion and exclusion criteria in the Case Series section as follows:
“Inclusion criteria were patients aged >20 and <80 years, with a clinical diagnosis of adhesive capsulitis or symptomatic rotator cuff tears, who had undergone at least 3 months of conservative therapy under orthopedic supervision without improvement, and with stable major organ function. Exclusion criteria included rheumatoid arthritis, neoplasm, severe arteriosclerosis, pregnancy, dementia, joint instability, iodine allergy, severe cardiac or cerebrovascular disease, neurological disease, uncontrolled diabetes mellitus, severe infection, bleeding tendency, and pain due to cervical or lumbar spine pathology.”

Comment 3. The text contains information from the article formatting template that needs to be removed (page 5, end of the discussion section).

Response 3: Thank you very much for pointing out our oversight. We have deleted that part.

Round 2

Reviewer 1 Report

Comments and Suggestions for Authors

Dear authors of paper jcm-3815954, I extend my warm congratulations for the improvements you made to your work. Your work is now easier to understand.